# AutomaTikZ: Text-Guided Synthesis of Scientific Vector Graphics with TikZ

**Jonas Belouadi**
Natural Language Learning Group
Bielefeld University, Germany
`jonas.belouadi@uni-bielefeld.de`

**Anne Lauscher**
Data Science Group
University of Hamburg, Germany
`anne.lauscher@uni-hamburg.de`

**Steffen Eger**
Natural Language Learning Group
University of Mannheim, Germany
`steffen.eger@uni-mannheim.de`

## Abstract

Generating bitmap graphics from text has gained considerable attention, yet for scientific figures, vector graphics are often preferred. Given that vector graphics are typically encoded using low-level graphics primitives, generating them directly is difficult. To address this, we propose the use of TikZ, a well-known abstract graphics language that can be compiled to vector graphics, as an intermediate representation of scientific figures. TikZ offers human-oriented, high-level commands, thereby facilitating conditional language modeling with any large language model. To this end, we introduce DaTikZ, the first large-scale TikZ dataset consisting of 120k TikZ drawings aligned with captions. We fine-tune LLaMA on DaTikZ, as well as our new model CLiMA, which augments LLaMA with multimodal CLIP embeddings. In both human and automatic evaluation, CLiMA and LLaMA outperform commercial GPT-4 and Claude 2 in terms of similarity to human-created figures, with CLiMA additionally improving text-image alignment. Our detailed analysis shows that all models generalize well and are not susceptible to memorization. GPT-4 and Claude 2, however, tend to generate more simplistic figures compared to both humans and our models. We make our framework, AutomaTikZ, along with model weights and datasets, publicly available.[1]

## 1 Introduction

Recent advancements in text-to-image generation have facilitated the generation of detailed images from simple natural language descriptions (Esser et al., 2021; Ramesh et al., 2021; 2022; Saharia et al., 2022; Rombach et al., 2022; Zhang et al., 2023a). Models like Stable Diffusion (Rombach et al., 2022) and DALL-E (Ramesh et al., 2021; 2022) often yield results comparable to real photographs or human-created artworks. However, these models primarily generate *raster graphics*, typically at low resolutions, which are not ideal for *scientific figures*. Researchers use scientific figures to convey complex ideas or present critical findings, making them central to scientific research (Tufte, 1992; Hsu et al., 2021). Consequently, they demand a high degree of geometric precision and legible text, even at small font sizes, areas where raster graphics fall short. As a result, many research conferences advocate the use of *vector graphics*,[2] which decompose information into geometric shapes, allow searchable text, and usually have smaller file sizes.

Automated vector graphics generation is a growing research area as well (Lopes et al., 2019; Carlier et al., 2020; Aoki & Aizawa, 2022; Ma et al., 2022; Frans et al., 2022; Jain et al., 2023; Wu et al., 2023), but current methods have their own share of limitations. Specifically, they mainly generate low-level path elements of the Scalable Vector Graphics (SVG) format, either failing to maintain

---

[1] https://github.com/potamides/AutomaTikZ
[2] https://acl-org.github.io/ACLPUB/formatting.html

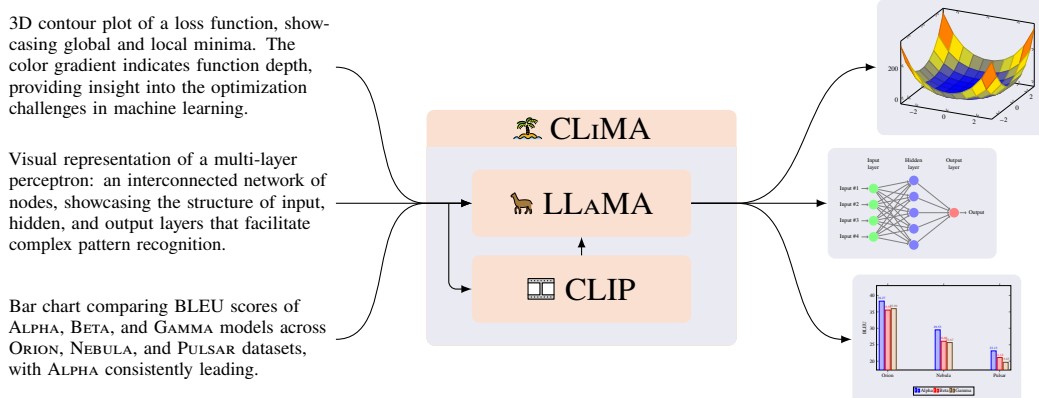

3D contour plot of a loss function, showcasing global and local minima. The color gradient indicates function depth, providing insight into the optimization challenges in machine learning.

Visual representation of a multi-layer perceptron: an interconnected network of nodes, showcasing the structure of input, hidden, and output layers that facilitate complex pattern recognition.

Bar chart comparing BLEU scores of ALPHA, BETA, and GAMMA models across ORION, NEBULA, and PULSAR datasets, with ALPHA consistently leading.

Figure 1: Exemplary scientific figures generated with CLiMA. CLiMA takes the captions as input, processes them with CLIP and LLaMA, and generates Ti*k*Z drawings that compile to vector graphics.

accurate geometric relations (Ma et al., 2022; Frans et al., 2022; Jain et al., 2023) or only generating outputs of limited complexity such as single icons or font characters (Lopes et al., 2019; Carlier et al., 2020; Aoki & Aizawa, 2022; Wu et al., 2023).

To address these limitations, we explore the use of *graphic languages*, which abstract from lower-level vector graphics formats by providing high-level constructs that can be compiled to such formats (Van Zandt, 2007; Hobby, 2014; Tantau, 2023). Language models show potential in learning these languages to solve simple tasks (Bubeck et al., 2023; Zhang et al., 2023b), but the depth of this capability, i.e., whether it can produce scientific figures, remains unexplored. Due to its expressiveness and emphasis on science, which enables the creation of complex figures with only a few commands, we focus on the graphics language *TikZ* in this work (Tantau, 2023). We aim to understand whether language models can capture the nuances of Ti*k*Z and automatically generate scientific figures based on image captions, analogous to text-to-image generation. This could not only enhance productivity and foster inclusiveness (aiding researchers less versed in programming-like languages, such as social scientists), but also aid education by creating tailored Ti*k*Z examples. The use case for this is demonstrated by the TEX Stack Exchange[3], where nearly 10% of the asked questions pertain to Ti*k*Z, making it the most frequently discussed topic on the platform. Our key contributions are as follows:

 (i) As part of our AUTOMATI*k*Z project, we create DATI*k*Z, the first large-scale Ti*k*Z dataset to our knowledge, featuring approximately 120k paired Ti*k*Z drawings and captions.

 (ii) We fine-tune the large language model (LLM) LLaMA (Touvron et al., 2023a) on DATI*k*Z and compare its performance to general-purpose LLMs, specifically GPT-4 (OpenAI, 2023) and CLAUDE 2 (Anthropic, 2023). Both automatic and human evaluation agree that scientific figures generated by fine-tuned LLaMA resemble human-created figures more closely.

(iii) We further develop CLiMA, a variant of LLaMA augmented with multimodal CLIP embeddings (cf. Figure 1; Radford et al., 2021). This enhancement allows CLiMA to visually interpret input captions, thereby improving text-image alignment. It also enables the use of images as supplementary inputs, leading to a further boost in performance.

(iv) In addition, we demonstrate that all models exhibit few memorization problems and generate novel outputs. However, GPT-4 and CLAUDE 2 tend to generate simpler outputs than LLaMA and CLiMA, sometimes producing degenerate solutions that maximize text-image similarity by visibly copying the input caption into the output image.

## 2 RELATED WORK

Our work connects to several distinct but interrelated fields, namely text-to-image generation, vector graphics generation, scientific figure understanding, and code generation. For each field, we provide a comprehensive review of the most relevant prior work.

---

[3]https://tex.stackexchange.com

**Text-to-Image & Vector Graphics Generation**   The evolution of text-to-image generation can be characterized by three development stages: Generative Adversarial Networks (Reed et al., 2016; Zhang et al., 2017; Brock et al., 2019; Kang et al., 2023), auto-regressive models (Ramesh et al., 2021; Ding et al., 2021; 2022; Chang et al., 2023), and diffusion models (Rombach et al., 2022; Ramesh et al., 2022; Saharia et al., 2022; Zhang et al., 2023a). Although Rodriguez et al. (2023a;b) explore their use for scientific figures, these approaches are inherently limited to generating raster graphics.

Vector graphics generation has evolved as a parallel field. Building upon innovative work in sketch generation (Ha & Eck, 2018), Lopes et al. (2019) generate single SVG font characters made up of straight lines and Bézier curves. Carlier et al. (2020) extend this approach to include SVG *icons*. However, none of these models support text conditioning. Another branch of research focuses on vectorizing text-to-image models (Ma et al., 2022; Frans et al., 2022; Jain et al., 2023). Although these methods enable text conditioning, text-to-image models typically have difficulties producing flat-colored SVG-style images, and the vectorized results tend to have imprecise geometric relations and jagged paths (Wu et al., 2023). Addressing these challenges, Cai et al. (2023) and Wu et al. (2023) investigate auto-regressive language modeling directly on SVG representations. Even though these approaches better capture the aesthetics of vector graphics, they are limited to editing existing graphics or generating monochrome icons of limited complexity.

**Scientific Figure Understanding**   Despite the limited number of approaches dedicated to generating scientific figures, scientific figure *understanding* is a subject of extensive research. Arguably, the task that is inverse to ours is the captioning of scientific figures. Expanding on prior work in scientific visual question-answering (Siegel et al., 2016; Kahou et al., 2018; Kafle et al., 2018), Chen et al. (2019a;b; 2020) train a captioning model using a corpus of synthesized figure-caption pairs. Hsu et al. (2021) extend this approach to real scientific figures, noting substantial challenges. To improve performance, Yang et al. (2023) reformulate the task, augmenting captions with OCR tokens and paragraphs referencing the figures. Singh et al. (2023) take a different approach and utilize reinforcement learning to consider the quality of the captions during training. In addition to such task-oriented models, recent advancements in multimodal large language modeling (MLLM; Liu et al., 2023; Dai et al., 2023; Yin et al., 2023) allow for generalized visual reasoning about scientific figures (Ye et al., 2023; Zhang et al., 2023c; Horawalavithana et al., 2023).

**Code Generation**   As graphics languages are a subset of programming languages, our work is closely related to code generation (Xu et al., 2022). At its core is the ongoing research on pre-training or fine-tuning LLMs on code (Rozière et al., 2023; Li et al., 2023; Fried et al., 2023; Li et al., 2022b; Chen et al., 2021), commonly with a multitask objective (Fried et al., 2023) consisting of causal language modeling and infilling prediction (Bavarian et al., 2022). Despite the significant amount of recent progress, the primary focus of code generation remains on high-resource programming languages such as Python, Java, or JavaScript (Zan et al., 2023). TikZ commands, in comparison, are invoked as TEX macros, and the TEX programming language is considered low-resource and typically overlooked in model evaluations. Yet, TEX may still exist in training corpora, as evidenced by GPT-4's ability to comprehend TEX and TikZ (Bubeck et al., 2023; Zhang et al., 2023b). As far as we know, there has been no comprehensive evaluation of this capability, which we also address in this work.

## 3   THE DATIkZ DATASET

DaTikZ is, to our best knowledge, the first large-scale dataset of TikZ drawings with corresponding captions. TikZ is well-known within the TEX community, but its resources are scattered across the internet, making the creation of a large dataset a fundamental challenge of our work. Consequently, we gather TikZ drawings and their captions from a variety of online resources, as detailed below.

### 3.1   DATA ACQUISITION

We collect the data from dedicated websites, online repositories, the TEX Stack Exchange, arXiv papers, and also artificial examples. A comprehensive overview of our data collection is provided in Table 1. We gather TikZ drawings created between November 2006 and June 2023 that successfully compile with TEX Live 2023.[4] Ablation studies and examples can be found in Appendices C and E.

---

[4]https://tug.org/texlive

**Curated Examples**  Several websites and on-line repositories[5] focus on collecting and sharing TikZ drawings for educational purposes and to showcase high-quality examples. Through web scraping, we retrieve any TikZ drawings from these sites that have associated captions.

| Source | Size | Augmented |
|---|---|---|
| Curated Examples | 981 | 63.2% |
| TEX Stack Exchange | 29 238 | 51.31% |
| ArXiv Papers | 85 656 | 67.75% |
| Artificial Examples | 3 914 | 50% |
| All | 119 789 | 62.71% |

Table 1: Detailed breakdown of DaTikZ showing size and percentage of augmented data for the whole dataset and each source individually.

**TEX Stack Exchange**  We also source TikZ drawings from TEX Stack Exchange (cf. §1). We examine the quarterly data dump and extract questions tagged with TikZ and relevant answers with a minimum score of 1. To convert textual questions into image captions, we utilize WizardLM (Xu et al., 2023), an LLM trained to follow arbitrary instructions. Using the title and body of a question as context, we task WizardLM with creating a descriptive caption for the figure provided in the answer. More details on the caption generation procedure can be found in Appendix B.

**ArXiv Papers**  ArXiv[6] is a widely-used open-access archive for scientific articles. As arXiv encourages researchers to upload their papers alongside their source files, it serves as a valuable resource for obtaining TikZ drawings. Initially, we retrieve all papers with TEX source files and retain those that use the TikZ package. Subsequently, we expand any include directives and extract all TikZ environments using regular expressions. To ensure compilability, we additionally preserve all required preamble elements. For that, we first establish a set of rules by analyzing documents obtained from other sources that determine which package imports and configuration options should be retained. We then parse all macro definitions and keep for each TikZ drawing the macros it uses. Finally, we exclude any TikZ drawings that fail to compile after this extraction process (around 120k).

**Artificial Examples**  GPT-4 has demonstrated the emergent ability to generate simple, tangible objects (e.g., unicorns) in TikZ (Bubeck et al., 2023). While not the primary focus of this work, we seek to transfer this ability to our models through knowledge distillation (Bucila et al., 2006). To this end, we compile a diverse set of object descriptions derived from the object categories in the MS COCO, LVIS, and VISOR datasets (Lin et al., 2014; Gupta et al., 2019; Gokhale et al., 2023). Moreover, we sample emoji descriptions from the OpenMoji database.[7] Following this, we instruct GPT-4 to generate a TikZ drawing for each description, using a chain-of-thought prompt (Wei et al., 2022) we adopt from Zhang et al. (2023b), as detailed in Appendix B.

## 3.2 Data Augmentation

Prior research indicates a correlation between caption length and caption quality (Gelman et al., 2002; Hartley, 2003; Huang et al., 2023). Notably, Huang et al. (2023) propose a heuristic to judge scientific captions with less than 30 tokens as being of poor quality. Given the recent advancements in MLLM, and notably in MLLM with a focus on science (cf. §2), we propose the automatic augmentation of such captions (Belouadi & Eger, 2023a;b). For an in-depth analysis of implications, refer to Appendix C.

Specifically, we leverage LLaVAR (Zhang et al., 2023c), instructing it to generate short descriptions for TikZ drawings with captions containing fewer than 30 tokens (cf. Appendix B).[8] Inspired by the CapFilt method (Li et al., 2022a), we generate five candidate descriptions and rank them based on their text-image similarity using CLIPScore (Hessel et al., 2021). The final augmented caption is then formed by concatenating the original caption with the top-ranked description. For GPT-4, we cannot rely on the heuristic, as the captions used are not scientific. Instead, we augment all captions to increase diversity in our dataset while retaining the original captions as well. Table 1 displays the percentage of augmented captions in our dataset. On average, this method increases the CLIPScore of captions with originally fewer than 30 tokens from 24.76 to 29.12, a substantial improvement in text-image similarity, especially considering that CLIPScore typically ranges between zero and 40 (Hessel et al., 2021). The CLIPScore for original captions exceeding 30 tokens is 27.06.

---

[5] https://texample.net, https://tikz.net, https://pgfplots.net & https://github.com projects
[6] https://arxiv.org
[7] https://openmoji.org
[8] In this work, we use the Moses tokenizer (Koehn et al., 2007) to count tokens.

## 4 METHODS

We leverage LLAMA (Touvron et al., 2023a) as the base model in most experiments, using captions from DATi*K*Z as model input and Ti*k*Z code as ground truths. Since TEX source files from arXiv were included in LLAMA's pre-training data, it may have prior knowledge beneficial for this task. We choose the original LLAMA release over its updated revisions, LLAMA 2 (Touvron et al., 2023b) and CODELLAMA (Rozière et al., 2023), as their training data is not as clearly specified. This uncertainty and their more recent release would make it difficult to create a test set without training-to-test data leakage. We also experiment with GPT-4 and CLAUDE 2, as earlier research hints at inherent potential for our task (cf. §3.1 and §2), and employ the same chain-of-thought approach outlined in §3.1. However, as they are proprietary, we can only address data leakage for LLAMA (Aiyappa et al., 2023).

### 4.1 CLIMA

A potential drawback of vanilla LLAMA is that it may not understand visual concepts, given that it was not designed to process image data. However, this ability could significantly enhance the creation of scientific figures. Therefore, we modify LLAMA by combining it with a CLIP ViT-H/14 model (Cherti et al., 2023). CLIP is frequently employed to establish a bridge between vision and natural language, thereby facilitating the creation of MLLMs (Yin et al., 2023).

However, unlike most MLLM methods, to our knowledge we are the first to use CLIP's *multimodal* projection layer, which allows us to extract visual information from both text and images in a common embedding space (cf. Figure 1). This approach is akin to text-to-image models like DALL-E and CLIP-GEN (Wang et al., 2022b), that make use of this duality to generate raster graphics. In our case, our primary objective is to provide LLAMA with a visual interpretation of the input caption, anticipating that this adjustment will boost the alignment with generated Ti*k*Z drawings. In addition, it also enables us to experiment with supplying rasterized scientific figures as an additional input (cf. §5). As this new model can be described as using **CLIP i**nside LL**A**MA, we refer to it as *CLIMA*.

We accomplish this integration by connecting CLIP's output with LLAMA's input via soft prompting (Lester et al., 2021); i.e., we prepend CLIP's embedding vector to LLAMA's input embeddings of the caption. This requires adding a feed-forward layer with dimensions $\delta_{\text{ViT-H/14}} \times \delta_{\text{LLAMA}}$ to connect image features of dimension $\delta_{\text{ViT-H/14}}$ with LLAMA's word embedding space of dimension $\delta_{\text{LLAMA}}$. Following insights from Liu et al. (2023), we pre-train this adaption layer on a dataset of 595K generic text-image pairs for one epoch while keeping both CLIP and LLAMA frozen during the process.

### 4.2 ERROR HANDLING & CORRECTION

A general issue with our language modeling approach to Ti*k*Z generation is that outputs may violate the syntactic and semantic rules of TEX, potentially leading to errors and uncompilable documents. While there are constrained decoding algorithms that can force models to form valid programs (Poesia et al., 2022; Scholak et al., 2021), they depend on parse trees and are only useful for languages with a context-free grammar. TEX, however, has a flat, unstructured syntax that is generally impossible to parse (Erdweg & Ostermann, 2010), rendering these methods unsuitable for our approach.

As an alternative, we propose an *iterative resampling* method, leveraging the diagnostic data produced during compilation. If an error arises during compilation, we analyze the logfile to identify its source. Rather than resampling from the start, we then reverse the generation process to just before the error line and continue sampling from there. If the error persists, we infer that the origin of the problem lies earlier in the code and reverse further back, specifically $4^{(i-1)}$ lines above the error, with $i$ denoting the current iteration. While this method does not guarantee error-free results, it provides a more efficient and targeted strategy than simply reinitiating sampling from the beginning.

## 5 EXPERIMENTS

Before fine-tuning our models on DATi*K*Z, we extract a sample of 1k *human-created* items to serve as our test set. As LLAMA's training started in December 2022, we only sample items introduced after this date to avoid data leakage. We conduct both automatic (§5.1) and human evaluations (§5.2). Additional results and instances of generated Ti*k*Z drawings are available in Appendices C and E.

**Model Sizes**    In terms of model size, we fine-tune LLaMA$_{7B}$ and CLiMA$_{7B}$, each with 7 billion parameters (7b), as well as LLaMA$_{13B}$ and CLiMA$_{13B}$ with 13 billion parameters (13b), respectively. During inference, we additionally evaluate CLiMA$_{13B}$ with CLIP receiving compiled human-created TikZ drawings as input instead of captions, which we refer to as CLiMA$_{IMG}$ for clarity (cf. §4.1).

**Training**    Given the size of these models, we introduce trainable low-rank adaption weights (LoRA; Hu et al., 2022) while keeping the base model weights frozen and in half precision (Micikevicius et al., 2018). Following Dettmers et al. (2023), we apply LoRA to all linear layers. In addition, we find that training the embedding layer and language modeling heads is crucial for successful fine-tuning. Since we are not aware of any studies applying LoRA to these layers, we make them fully trainable and leave this investigation to future work. In line with Liu et al. (2023), we train for 12 epochs with AdamW (Loshchilov & Hutter, 2019) and a batch size of 128, but increase the learning rate to 5e−4 as this leads to faster convergence. As a form of data augmentation only possible for CLiMA, we randomly replace the captions forwarded to CLIP with the reference image in 50% of the cases.

## 5.1    Automatic Evaluation

We use a variety of automatic evaluation metrics to evaluate the performance of our models on our test set in terms of code, image, and caption-image similarity. In particular, we use the following metrics:

**CLIPScore**    calculates the similarity between image and text, as outlined in §3.2. We utilize it to evaluate the correlation between a rasterized TikZ drawing and its corresponding caption.

**CLIPScore$_{IMG}$**    is technically the same metric as CLIPScore, but with human-made TikZ drawings as a reference input. Therefore, it assesses the similarity of two images rather than an image and a caption. To our best knowledge, we are the first to use CLIPScore in this configuration.

**Kernel Inception Distance (KID)**    assesses the quality of generated TikZ drawings by comparing their distribution with the distribution of real images in the test set (Binkowski et al., 2018). This comparison helps determine how realistic the generated images appear in general. We extract image features using the same CLIP model utilized in CLIPScore.

**CrystalBLEU**    is an n-gram-based metric designed to measure textual similarity (Eghbali & Pradel, 2023). As a variant of BLEU (Papineni et al., 2002), optimized for evaluating code, we employ it to assess the similarity between human-created and machine-produced TikZ code.

**Extended Edit Distance (EED)**    is a metric dedicated to assessing string similarity (Stanchev et al., 2019), much like CrystalBLEU. We utilize it to determine the minimum number of operations needed to convert the machine-generated code into the reference code.

**Compilation Sampling Rate (CSR)**    measures how frequently we need to sample from a model to yield compilable TikZ code that outputs an image. This is crucial as some metrics depend on images. With LLaMA and CLiMA, we use iterative resampling (cf. §4.2) and account for partial samples. This is not feasible with GPT-4 and Claude 2 due to their chain-of-thought prompt, which generates code across multiple steps. We take a relaxed stance, counting a sample as successful if it results in an image, even if there are errors.

**Results**    We compute the above metrics for each model and present the system-level scores in Figure 2. The radar chart on the left illustrates that there are small but noticeable score differences between the LLaMA and CLiMA models, revealing some intriguing trends. Aligning with the intuitive expectation that larger models yield better performance (Kaplan et al., 2020), the 13b models clearly outperform the 7b models on all string-similarity and image-based metrics by 0.2–0.5pp (percentage points). A notable exception is CSR, where all models perform comparably. This shows that all models require approximately 1.2 samples per caption to generate a compilable TikZ drawing.

Within model sizes, CLiMA$_{7B}$ outperforms LLaMA$_{7B}$ on CrystalBLEU, EED, CLIPScore, and CLIPScore$_{IMG}$ by up to 0.4pp, suggesting that, even when only text inputs are involved, integrating CLIP into the model has a predominantly positive effect. CLiMA$_{13B}$ continues this trend, showing a 0.1pp higher CLIPScore than LLaMA$_{13B}$. However, we also see that this does not necessarily have to increase the similarity with a reference image as well, as LLaMA$_{13B}$ has a 0.1pp higher CLIPScore$_{IMG}$. On CrystalBLEU and EED, CLiMA$_{13B}$ again fares better, although, with 0.1pp, the gap is not as pronounced as for the 7b models, possibly due to diminishing returns (Hong et al., 2023).

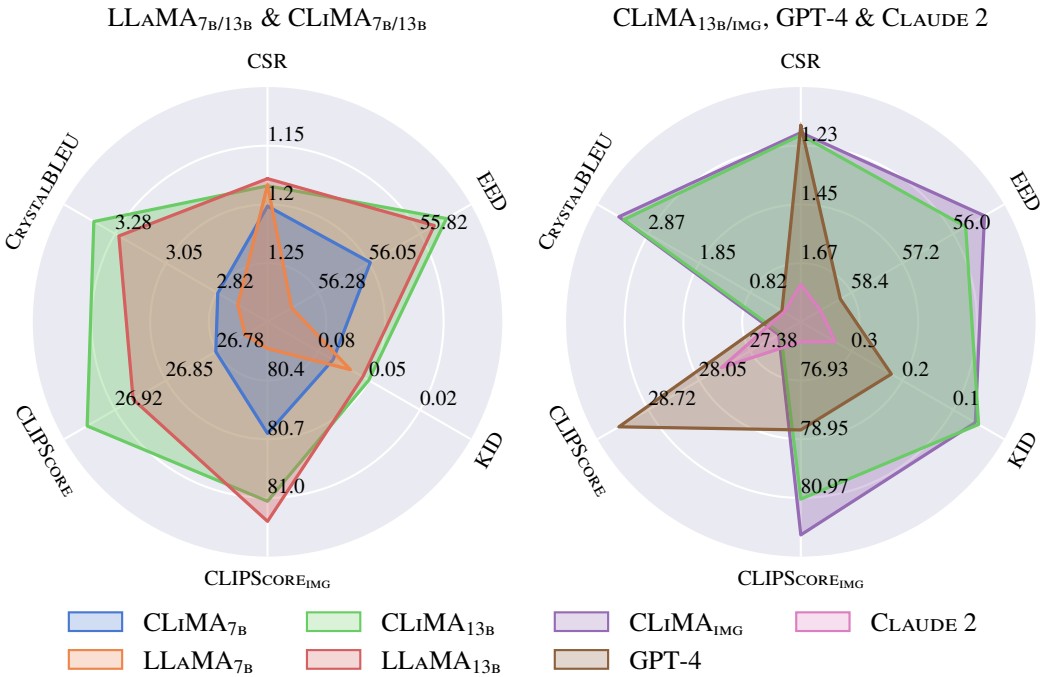

Figure 2: Automatic evaluation results for LLaMA_{7B/13B}, CLiMA_{7B/13B/IMG}, GPT-4, and Claude 2. Axes representing metrics where lower values are better (CSR, EED, and KID) have been inverted. Detailed scores are provided in Appendix C for further reference.

The right radar chart compares our best text-only model, CLiMA_{13B}, with CLiMA_{IMG}, GPT-4, and Claude 2. As before, all models perform roughly the same on CSR, except for Claude 2, which needs noticeably more samples. As expected, CLiMA_{IMG}, having access to reference images, improves upon CLiMA_{13B} in CLIPScore_{IMG} by 1.2pp. However, this does not lead to an improvement in CLIPScore, echoing our earlier observation that image and caption-image similarity do not always correlate. It also does not improve KID, demonstrating that the overall quality of the images remains constant. Nevertheless, the string-based metrics are 0.1–0.4pp higher, indicating that conditioning on reference images positively impacts code similarity.

We also observe that Claude 2 performs much worse than GPT-4, and both perform noticeably worse than both CLiMA_{13B} and CLiMA_{IMG} on most metrics. The drastically lower CrystalBLEU and EED (up to 3.9pp) suggest that GPT-4 and Claude 2 generate fundamentally different code (in Appendix A we show that it exhibits a lower level of complexity). The up to 6.6pp lower CLIPScore_{IMG} and over six times as large KID indicate that not only do the generated images look different from human ones, but also that the general quality of images is much different from the human distribution. However, most strikingly, both models achieve an up to 2.1pp higher CLIPScore. Upon investigation, we find that both models tend to produce degenerate images, which visibly copy the input caption into the output image. Since the outputs of CLIP (and by extension CLIPScore) can be controlled with *images of text* (Ramesh et al., 2022), Claude 2, and in particular GPT-4, essentially employ such typographic attacks to achieve exceptional caption-image similarities. We further explore this phenomenon in §6.

Overall, CLiMA_{7B} and CLiMA_{13B} outperform their respective LLaMA models in five out of seven metrics each, with Claude 2 and GPT-4 substantially underperforming all of them. While CLiMA_{IMG} unsurprisingly improves upon CLiMA_{13B}, CLiMA_{13B} is the best model with only textual inputs.

## 5.2 Human Evaluation

To further evaluate the effectiveness of our models, we conduct a human annotation campaign using *best-worst scaling* (BWS; Louviere et al., 2015). As a form of comparative annotation, BWS yields high-quality results even when the number of annotations is low (Kiritchenko & Mohammad, 2016;

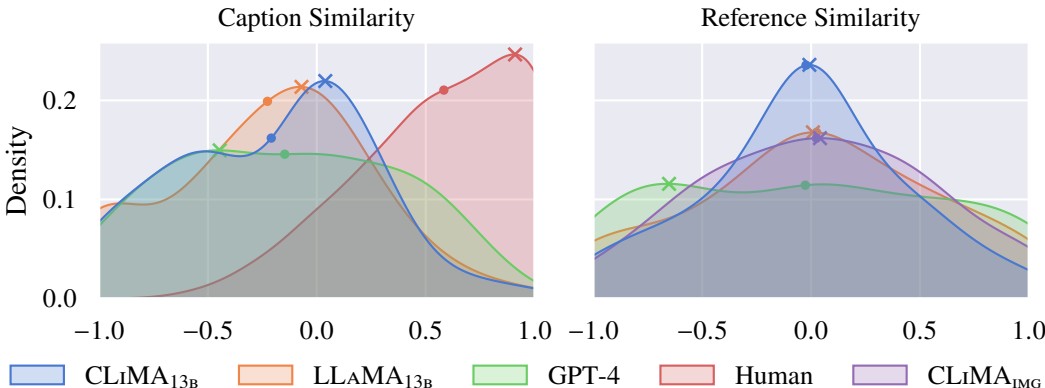

Figure 3: Distributions of BWS scores per model for caption and reference similarity. Scores span from -1 (poor) to 1 (excellent). The "•" markers denote expected values, and "×" signifies the mode.

2017). Within this method, annotators are tasked to compare tuples of $n = 4$ items, identifying the best and the worst item based on a given property. Real-valued scores, ranging from -1 (poor) to 1 (excellent), are then computed by subtracting the fraction of times an item is chosen as the worst from the fraction of times it is chosen as the best (Orme, 2009).

In this work, we focus on *caption similarity* (CS) and *reference similarity* (RS). In CS, annotators evaluate image tuples based on text-image similarity with captions (similar to CLIPSCORE). We construct 4-tuples consisting of our two leading text-only models from automatic evaluation (CLIMA$_{13B}$ and LLAMA$_{13B}$), GPT-4, and human reference images. In RS, the human reference images are used as the standard of comparison instead (similar to CLIPSCORE$_{IMG}$), so we replace them in the tuples with CLIMA$_{IMG}$, while leaving the other models unchanged. Each property is then annotated by four unique expert annotators with relevant research experience (cf. Appendix D).[9] To ensure a manageable workload for the annotators, we create our tuples from a subset of 100 items sampled from our test set. We assess the consistency of the annotators using *split-half reliability* (SHR; Kiritchenko & Mohammad, 2017). This method involves randomly splitting all annotations into two sets, calculating scores for each set, and then determining the correlation between them using Spearman's $\rho$.

**Results** For CS, the SHR is $\rho = 0.6$, indicating a moderate but adequate consensus among annotators. Figure 3 (left) exhibits kernel density estimates for the computed scores, with marked modes and expected values. Unsurprisingly, humans perform best with a mode near 1. CLIMA$_{13B}$ is the only other model with a mode above 0, followed by LLAMA$_{13B}$, while GPT-4 lags behind. This indicates that when sampling once with a given caption, CLIMA$_{13B}$ is most likely to generate the best image. Since CLIMA$_{13B}$ and LLAMA$_{13B}$ retain their earlier CLIPSCORE ranking, but GPT-4 drops substantially, we hypothesize that human annotators are not as prone to typographic attacks as metrics. However, we still observe a slight bias towards images of text. In 75% of cases where GPT-4 is selected as the best model, it copies more n-grams from the caption into the image than the worst-ranked image, potentially leading to outliers and thus a slightly higher expected value than CLIMA$_{13B}$ or LLAMA$_{13B}$.

Regarding RS, we record a similar SHR, with $\rho = 0.58$. For LLAMA$_{13B}$, CLIMA$_{13B}$, and CLIMA$_{IMG}$, the distributions in Figure 3 (right) are almost normally distributed, with the mode and expected value being nearly identical. As with CLIPSCORE$_{IMG}$, LLAMA$_{13B}$ is ranked marginally higher than CLIMA$_{13B}$, indicating CLIPSCORE$_{IMG}$ correlates well with human rankings (Appendix C details exact correlations). On a similar scale, CLIMA$_{IMG}$ outperforms LLAMA$_{13B}$. In contrast, GPT-4 follows a nearly uniform distribution, with a slight downward trend for better scores. Therefore, its mode is noticeably lower than for the other models. The expected value, albeit only slightly, is also the lowest.

In summary, our human evaluation aligns well with our automatic metrics, with the added benefit of lower susceptibility to typographic attacks. CLIMA$_{13B}$ outperforms LLAMA$_{13B}$ on CS, while CLIMA$_{IMG}$ surpasses LLAMA$_{13B}$ on RS. GPT-4 shows peculiar distributions, with the mode (and also the expected value for RS) lagging behind, highlighting the effectiveness of our models.

---

[9]We tried crowdsourcing as well, but due to low agreement with experts, we concluded that crowdworkers lack the necessary expertise for our tasks (cf. Appendix D).

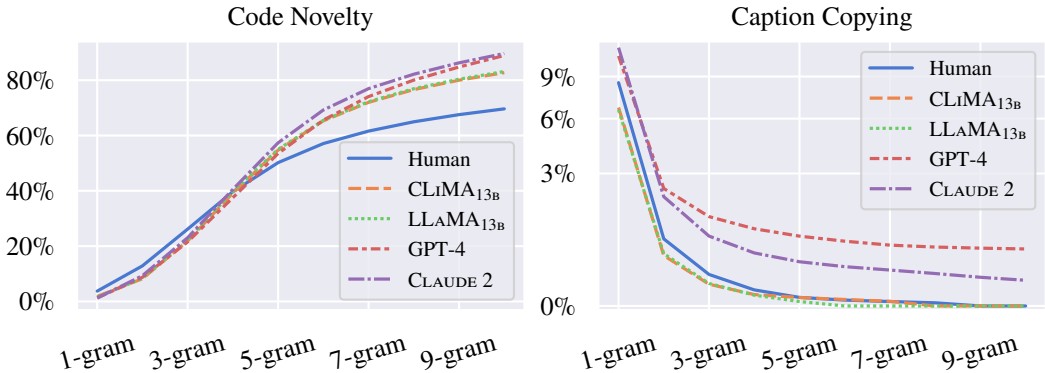

Figure 4: Proportion of unique code n-grams ($n \in [1, 10]$) that do not appear in the training data (left), and proportion of caption n-grams that were copied into the output image (right).

## 6 Analysis

The issue of language models memorizing and copying training data is a prevalent concern (McCoy et al., 2023; Carlini et al., 2023; Raunak & Menezes, 2022; Meehan et al., 2020). Similarly, we discovered in §5.1 that GPT-4 and Claude 2 tend to perform typographic attacks by memorizing and copying input captions. In this section, we analyze the extent of these issues on our test set using the concept of *n-gram novelty* (McCoy et al., 2023). Specifically, to measure *code novelty*, we determine the proportion of n-grams in the model-generated TikZ code that are *not* found in the training data. To measure *caption copying*, we calculate the proportion of n-grams from the caption that were copied verbatim into the output code. For comparison, we also calculate both metrics for human references.

Figure 4 displays the results for both metrics ($n \in [1, 10]$) after filtering code comments. In terms of code novelty, models tend to generate less novel code than humans for smaller n-grams ($n < 4$). However, for $n > 6$, models become more novel, with more than 80% of all model n-grams being novel for $n > 8$. McCoy et al. (2023) observe the same phenomenon in all datasets investigated and conclude that this ratio is the normal case when a model is not affected by memorization of the training data. Regarding caption copying, GPT-4 and Claude 2 copy considerably more n-grams from the caption than our models. For 1-grams (i.e., $n = 1$), CLiMA₁₃ᵦ and LLaMA₁₃ᵦ copy around 6.5% of n-grams, while GPT-4 and Claude 2 copy more than 10%. For $n > 5$, CLiMA₁₃ᵦ, LLaMA₁₃ᵦ, and humans practically stop copying, but Claude 2 and especially GPT-4 continue with an almost linear trend, hinting at instances where they might copy the entire caption (cf. Appendix E for examples). This reinforces our hypothesis from §5.1 and points towards CLIPSCORE_IMG as a more robust metric for assessing the visuals of text-rich images since it seems less susceptible to typographic attacks.

## 7 Conclusion & Future Work

In this work, we present AutomaTikZ, a project for automatically generating TikZ drawings based on natural language descriptions. As part of AutomaTikZ, we release DaTikZ, a pioneering dataset of aligned TikZ drawings and captions, and CLiMA, a novel model architecture which integrates multimodal CLIP embeddings into LLaMA. By fine-tuning CLiMA on DaTikZ, we demonstrate that it outperforms LLaMA on several metrics and also surpasses proprietary GPT-4 and Claude 2. In addition, CLiMA can also process images, potentially extending its application to vectorization and conditioning on sketches. Important finds are that (i) integrating CLIP can lead to improvements, even when only text inputs are involved, provided the task relates to visual concepts, and that (ii) attention should be paid to typographic attacks when evaluating models that generate text-rich images.

In future research, we aim to incorporate insights from the caption generation community and enrich our input texts with other figure-mentioning sections of the source documents (cf. §2). We also plan to enhance our extraction pipeline, especially since we had to exclude over 120k TikZ images from arXiv that failed to compile. We hope that these modifications will bring us a step closer to bridging the gap to human performance.

## 8 ETHICS STATEMENT

We ensure that the TikZ drawings we gather from online sources are licensed in a manner that permits us to copy and redistribute them. Most sources license their content under a Creative Commons attribution license,[10] the GNU Free Documentation License,[11] or the MIT license.[12] ArXiv is an exception in that, even though it allows licensing under a Creative Commons license, the majority of papers are published under a non-exclusive license, which does not grant us permission to redistribute.[13] As a result, we exclude any TikZ drawings from arXiv that use this license in the public release of DaTikZ. Nevertheless, we do release AutomaTikZ in conjunction with the dataset generation code, enabling anyone to recreate the full version of DaTikZ themselves. As for auto-generated samples, OpenAI prohibits the use of GPT-4 for creating competing services, restricting this part of our dataset to non-commercial applications.[14]

Apart from our dataset, in this work we compare openly developed models with the proprietary GPT-4 and CLAUDE 2, whose full training details and hyperparameters have not been published. While we strive for a fair evaluation of all models, the lack of transparency in proprietary systems inevitably hinders comparative evaluations and reproducibility.

Within the scope of these limiting aspects, however, our models perform best in generating TikZ across the tested conditions. Yet, our models should not be used as a substitute for human judgment and critical thinking. They may carry any biases, flaws, or gaps that exist in the base models and training data and could potentially misinterpret the input, fabricate non-existent details, or overlook crucial information. Users should be aware of potential differences between the results they expect and the output the models generate.

Furthermore, while our models are designed to aid in the production of legitimate scientific figures, they could potentially be used to generate disinformation and fake science in the hands of malicious actors.

## ACKNOWLEDGMENTS

We gratefully thank, in no particular order, Timm Dill, Yanran Chen, Daniil Larionov, JiWoo Kim, Vivian Fresen, Martin Kerscher, Christoph Leiter, and Ran Zhang for their help with our human evaluation campaign, proofreading, discussions, and comments on our work. We further thank the BMBF for its support via the grant METRICS4NLG and the Ministry of Culture and Science of the State of North Rhine-Westphalia for the grant no. NW21–059A (SAIL). The second author is funded under the Excellence Strategy of the German Federal Government and the Länder. The last author is supported by DFG grant EG 375/5–1. We also thank Hugging Face for providing a community GPU grant. Any icons used in this work were designed by OpenMoji, the open-source emoji and icon project.

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

## A  CODE & IMAGE COMPLEXITY

An interesting phenomenon we observe in our experiments in §5 is noticeable differences in TikZ code size. As shown in Table 2, human-created TikZ drawings, after filtering comments, contain twice the number of tokens as CLiMA$_{13B}$ and LLaMA$_{13B}$, which in turn have twice as many tokens as GPT-4 and Claude 2. We investigate how this discrepancy is reflected in the compiled images, in particular, whether images created from code with fewer tokens are less complex and whether this also affects image quality. To this matter, we gather another round of BWS scores. Analogous to §5.2, we form tuples with CLiMA$_{13B}$, LLaMA$_{13B}$, GPT-4, and humans. We assign a single annotator to evaluate image tuples in terms of

| Source | Tokens | Cplx | Qlty |
|---|---|---|---|
| Human | 916.56 | 0.76 | 0.83 |
| CLiMA$_{13B}$ | 422.56 | 0.46 | 0.4 |
| LLaMA$_{13B}$ | 420.28 | 0.46 | 0.39 |
| GPT-4 | 186.73 | 0.31 | 0.38 |
| Claude 2 | 179.42 | — | — |

Table 2: Average number of tokens in TikZ documents, along with averaged min-max normalized BWS scores for image complexity (Cmplx) and quality (Qlty).

complexity and, due to the potentially subjective nature of the task, four annotators to evaluate image quality. Since we obtain an SHR of $\rho = 0.68$ for image quality, we conclude that all annotators have a similar idea of what constitutes it.

We display averaged and min-max normalized BWS scores in Table 2. Unsurprisingly, there is a clear correlation between code size and image complexity. Humans score over twice as high as GPT-4, with CLiMA$_{13B}$ and LLaMA$_{13B}$ falling in between. A link to quality is also visible, although less pronounced. Humans excel distinctly, but CLiMA$_{13B}$, LLaMA$_{13B}$, and GPT-4 are not far apart. Nonetheless, in absolute terms, CLiMA$_{13B}$ still ranks higher than LLaMA$_{13B}$, and GPT-4 comes in last. Overall, this confirms our observations in §5.1 that GPT-4 and Claude 2 produce code most different from humans, yielding figures that are simpler than other models. Consequently, we believe that our tool can assist humans better in creating more detailed TikZ drawings.

## B  PROMPT ENGINEERING

Our work involves developing a series of prompts to guide general-purpose language models in accomplishing specific tasks. In this section, we discuss each model and the corresponding prompts we designed. Within a prompt, terms enclosed in curly braces symbolize placeholder variables that are substituted during inference.

**WIZARDLM**  We employ WizardLM to create captions for TeX Stack Exchange content based on a question and its title (cf. §3.1). We force the generated tokens to start with "Desired outcome:" which we subsequently strip from the output, along with any text generated after the first newline:

```
1  Create a clear and specific caption for an image that depicts the
2  desired outcome of the following question. Utilize all relevant
3  details provided in the question, particularly focusing on the
4  visual aspects.  Avoid referencing any example images or code
5  snippets. Ensure that your caption is comprehensive and accurate:
6
7  {TITLE}
8
9  {QUESTION}
```

**LLaVAR**  We use LLaVAR to generate short descriptions for scientific figures (cf. §3.2). We initially tried to provide LLaVAR with the original caption as context to improve it directly, but in many cases this seemed to confuse the model. Therefore, we decided to create short descriptions based solely on the figure and later append them to the original caption. We found that short, concise prompts work best for generating these descriptions:

```
1  {IMAGE}
2  Write a short description for the image.
```

| Model | CER$_\downarrow$ | CSR$_\downarrow$ | EED$_\downarrow$ | KID$_\downarrow$ | CLIP$_{\text{IMG}\uparrow}$ | CLIP$_\uparrow$ | BLEU$_\uparrow$ |
|---|---|---|---|---|---|---|---|
| LLaMA$_{7B}$ | 0.644 | 1.183 | 56.393 | 0.059 | 80.237 | 26.733 | 2.732 |
| LLaMA$_{13B}$ | 0.552 | 1.178 | 55.762 | 0.053 | 81.12 | 26.898 | 3.258 |
| CLiMA$_{7B}$ | 0.545 | 1.202 | 56.045 | 0.068 | 80.671 | 26.776 | 2.82 |
| CLiMA$_{13B}$ | 0.656 | 1.184 | 55.708 | **0.05** | 81.017 | 26.966 | 3.369 |
| CLiMA$_{\text{IMG}}$ | 0.658 | 1.175 | **55.264** | 0.057 | **82.252** | 26.994 | **3.469** |
| Claude 2 | 1.115 | 1.758 | 59.135 | 0.333 | 75.589 | 27.753 | 0.137 |
| GPT-4 | **0.384** | **1.147** | 58.667 | 0.222 | 78.63 | **29.115** | 0.181 |

Table 3: Precise system-level scores for automatic evaluation metrics. CLIPScore is abbreviated with CLIP and CrystalBLEU with BLEU. CER shows the average number of compile time errors. Arrows indicate whether larger or smaller scores are better, and the best scores are visually highlighted.

**GPT-4 & Claude 2**  To generate TikZ drawings (cf. §3.1, §4, and §5), we use the same prompt for GPT-4 and Claude 2, which was originally designed by Zhang et al. (2023b) and slightly modified for our task. After a comparison with our own prompts and prompts from other works (Bubeck et al., 2023), we found it to work best in our initial experiments. As a chain-of-thought prompt it breaks down the task of code generation into a series of logical steps. By default, the models tend to copy the input caption into the output image. To counter this, we additionally instruct the models not to include the caption in generated images. However, instances of caption copying still occur, as discussed in §6:

```
1  Draw a TikZ picture for the following caption: {CAPTION}. First,
2  you need to provide an abstract step-by-step drawing guide. Then,
3  you need to generate the full code (beginning with "\documentclass"
4  and ending with "\end{document}") following the guide. Avoid any
5  direct inclusion of the caption or any lengthy text sequences within
6  the code. Finally, summarize the drawing.
```

## C  Additional Details on Experiments

In this section, we conduct additional ablation studies and provide more specific aspects of our experiments in §5. This will provide a clearer understanding of our methodologies and their respective considerations and limitations.

### C.1  Training Challenges

During the training phase, we limit the context size of our models to 1200 tokens due to constraints of our existing GPU resources, and filter out all examples that exceed this limit. Although most examples were within these limits, they were generally very close to the maximum allowable length. This prevented us from training our models to perform functions such as multi-turn conversations like chatbots, which could have allowed compiler logs to be fed back into the model to obtain improved outputs. Although we did consider this approach in preliminary experiments with vanilla LLaMA and its instruction-tuned derivatives in a zero-shot setting, the baseline performance of these models was too poor to produce usable TikZ output. Instead, by drawing parallels to text-to-image generation models that condition the output solely on the input caption and using our iterative resampling technique to improve the outputs, we provide more resource-efficient alternatives with good performance.

### C.2  In-depth Evaluation Metric Scores

Table 3 presents the precise system-level scores from our automatic evaluation in §5.1. As observed in that section, CLiMA$_{\text{IMG}}$ and CLiMA$_{13B}$ generally outperform the other models, with CLIPScore being a notable exception due to GPT-4 and Claude 2 being prone to caption copying. The table also contains the Code Error Rate (CER), which indicates the average number of compile time errors of our models. As evidenced, it largely reflects the trends of CSR. Most models generate less than

| Training Configuration | CER↓ | CSR↓ | EED↓ | KID↓ | CLIP$_{IMG}$↑ | CLIP↑ | BLEU↑ |
|---|---|---|---|---|---|---|---|
| Full Training | 0.545 | 1.202 | 56.045 | 0.068 | 80.671 | 26.776 | 2.82 |
| −Caption Augmentation | 0.603 | 1.169 | 56.384 | 0.056 | 78.312 | 25.703 | 2.714 |
| −Image Training | 0.681 | 1.178 | 56.336 | 0.08 | 80.129 | 26.604 | 2.736 |
| −ArXiv Papers (86k) | 2.145 | 1.546 | 60.552 | 0.229 | 75.15 | 25.328 | 0.886 |
| −TEX Stack Exchange (29k) | 0.624 | 1.207 | 56.274 | 0.071 | 80.461 | 26.958 | 2.936 |
| −Artificial Examples (4k) | 0.503 | 1.194 | 56.51 | 0.079 | 79.912 | 26.68 | 2.537 |
| −Curated Examples (1k) | 0.656 | 1.22 | 56.246 | 0.072 | 80.346 | 26.815 | 2.711 |

Table 4: Ablation study results for CLiMA$_{7B}$ demonstrating how omitting either caption augmentation or randomly forwarding images to CLIP instead of captions during training impacts performance on the test set. The study also evaluates the impact of each data source by training CLiMA$_{7B}$ on subsets of DaTi$\kappa$Z, omitting one data source each time. Scores lower or higher than those from full training (taken from Table 3) are shaded in red and green, respectively. Color intensity mirrors each score's proportion to the minimum and maximum values of its metric.

one error on average, with CLAUDE 2 being the only exception. Overall, GPT-4 achieves the best score, possibly because it has fewer opportunities to make mistakes by generating shorter and less complex code than other models (cf. §A). Since CLIPScore, at a fundamental level, evaluates similar aspects to CS from our human evaluation in §5.2 (caption-image similarity), and CLIPScore$_{IMG}$ is the intuitive counterpart to RS (image similarity), we also calculate their respective Spearman correlations. At the segment-level, the correlation coefficients are 0.23 for CLIPScore and CS, and 0.25 for CLIPScore$_{IMG}$ and RS (the degrees of correlation aligning with values commonly found for machine translation metrics; Freitag et al., 2021; Rei et al., 2022). The system-level correlations are 0.5 and 0.8, respectively. The higher correlations witnessed for image over caption-image similarity lends credibility to our hypothesis regarding CLIPScore's biases favoring images of text.

## C.3 ABLATION STUDIES

In this subsection we delve into ablation studies performed to understand the degree to which data augmentation and the different data sources of DaTi$\kappa$Z contribute to the test set performance of our models.

**Impact of Data Augmentation** In our work, we incorporate two key data augmentation techniques: we (i) automatically augment image captions in DaTi$\kappa$Z (cf. §3.2), and we (ii) randomly replace image captions forwarded to CLIP with reference images during the training of CLiMA (cf. §5). Even though we show that caption augmentation improves caption-image alignment, we have yet to conclusively quantify its effect on model performance. Similarly, although mixing text-only and text-image modalities is common when training models accepting both unimodal and multimodal input (Li et al., 2020; Wang et al., 2022a), the implications of our particular training strategy remain to be explicitly measured. To investigate these matters, we conduct an ablation study using CLiMA$_{7B}$ as a representative model for efficiency reasons. In particular, we construct a version of CLiMA$_{7B}$ that uses DaTi$\kappa$Z without caption augmentation and another version where we train without sampling images. The performance on our evaluation metrics compared to full training results of CLiMA$_{7B}$ can be seen in Table 4.

Although eliminating caption augmentation reveals subtle improvements for CSR and KID, the model noticeably underperforms for all other metrics, particularly CLIPScore (1.1pp lower) and CLIPScore$_{IMG}$ (2.4pp lower). Interestingly, the metrics that show an improvement do not compare model outputs with individual references but either do not use references at all (CSR) or use them only to evaluate general characteristics (KID). On the other hand, metrics with worse results are mostly reference-based. We attribute this observation to the reduced amount of information carried by non-augmented captions, most likely leading to a decline in CLiMA's ability to capture the connection between captions and images and, therefore, delivering lower performance in reference-based metrics. Conversely, with less information that might otherwise constrain the output space, the model is likely to gain more flexibility in producing figures. We conjecture that this allows the model to concentrate

more on aspects that contribute positively to metrics without direct references, such as CSR and KID. This hypothesis provides a plausible explanation for our observed results.

Upon examining the impacts of not sampling images during the training process, we find that most metrics exhibit worse performance, with only CSR yielding comparable results. These findings support our choice of training strategy and show that integrating images throughout the training process can bolster performance even when only textual inputs are considered at the evaluation stage.

In summary, we observe that both data augmentation techniques contribute positively to model performance. Caption augmentation enhances the caption-image alignment, benefiting CLiMA$_{7B}$, while leveraging images during training results in improvements as well.

**Importance of each Data Source**   Given the diverse data sources making up DaTi$\kappa$Z (arXiv, TEX Stack Exchange, curated examples, and artificial examples; cf. §3.1), we aim to assess the contribution of each source to the test set performance. This exploration serves multiple purposes: firstly, it enables us to justify the integration of each source; secondly, it can inform future data collection initiatives. To realize this objective, we develop four additional variants of CLiMA$_{7B}$, where each variant is trained by omitting one data source and utilizing only the remaining three. As before, we plot the difference in performance compared to the full training results in Table 4, which should expose the importance of each data source.

The greatest performance decline occurs with the removal of arXiv, a consistent outcome across all metrics. As the primary contributor of scientific figures totaling 86k, its absence instigates a decline of 4.5pp for EED, 5.5pp for CLIPScore$_{IMG}$, 1.4pp for CLIPScore, and 1.9pp for CrystalBLEU. Similarly, CER inflates nearly fourfold, and CSR experiences a noticeable increase, as well. Strikingly, however, results diverge for DaTi$\kappa$Z's second largest source, the TEX Stack Exchange, which provides 29k examples. Contrary to expectations, the general downward trend is comparatively small, with EED and CLIPScore$_{IMG}$ only suffering a 0.2pp decrease and CLIPScore and CrystalBLEU even improving by 0.2pp and 0.1pp, respectively. We attribute this outcome to TEX Stack Exchange's unique focus on providing problem-specific minimum working examples rather than scientific figures like the rest of DaTi$\kappa$Z. Although this could still provide valuable training signals, we did not design our test set (only ~100 of its 1k random samples originate from this source) to evaluate the ability of our models to follow technical advice given in the figure captions. Consequently, we plan to improve our testing framework in future work. Next, although we exclude artificial examples from our test set due to their non-human origin and their removal from the training data improves CER and CSR, their absence still has a negative impact on all other metrics. Without artificial examples, EED drops by 0.5pp, CLIPScore$_{IMG}$ by 0.8pp, CLIPScore by 0.1pp and CrystalBLEU by 0.3pp, showing that distillation of GPT-4 is a useful component in the training process. Lastly, even though curated examples constitute the smallest data source (less than 1k examples), their omission from the training data induces greater negative performance impacts than the removal of TEX Stack Exchange (0.2pp for EED, 0.3pp for CLIPScore$_{IMG}$ and 0.1pp for CrystalBLEU). We attribute this disproportionate value per size to the likely higher quality of code and captions inherent to curated examples.

Overall, if the results are examined independently, we observe that each source has primarily positive effects. However, when evaluated in the context of the source sizes, we find that curated examples create a particularly noticeably positive impact, whereas TEX Stack Exchange exhibits smaller effects. These insights will guide us to focus more on collecting such curated examples in future work.

# D   ANNOTATOR DEMOGRAPHICS

Our annotators are proficient in English, possessing a C1 level or above as per the Common European Framework of Reference for Languages.[15] They all come from a science and technology background with research experience. They specifically consist of one female and one male faculty member, one female PhD student, three male PhD students, and two female and two male assistants from other institutions.

In addition to these expert annotators, we also conducted preliminary tests with crowdworkers via Amazon Mechanical Turk,[16] gathering ten crowd-annotations per tuple for each task. However, the

---

[15] https://www.coe.int/lang-cefr
[16] https://www.mturk.com

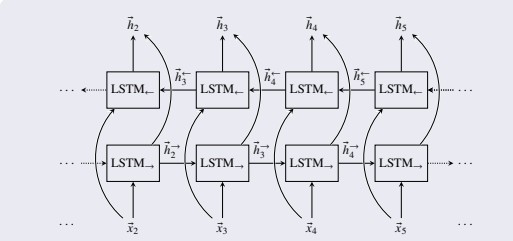

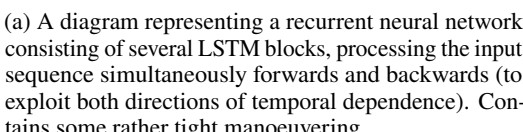

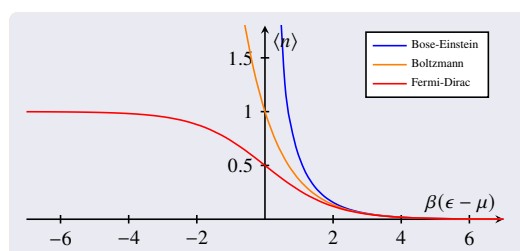

(a) A diagram representing a recurrent neural network consisting of several LSTM blocks, processing the input sequence simultaneously forwards and backwards (to exploit both directions of temporal dependence). Contains some rather tight manoeuvering.

(b) A plot comparing the distribution functions of Bose-Einstein, Boltzmann, and Fermi-Dirac statistics as a function of the reduced chemical potential $\beta(\epsilon - \mu)$. This visualiation highlights the differences between the three types of distribution functions, which are used to describe the behavior of particles in different statistical systems.

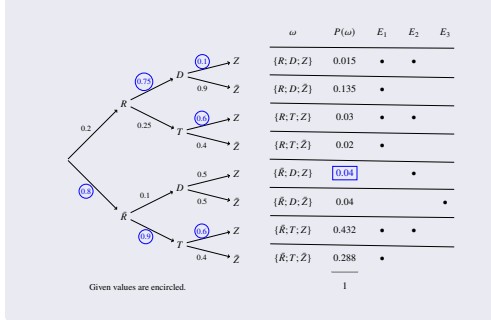

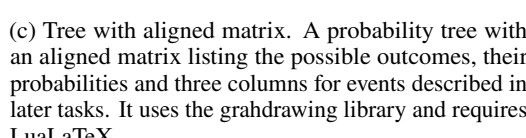

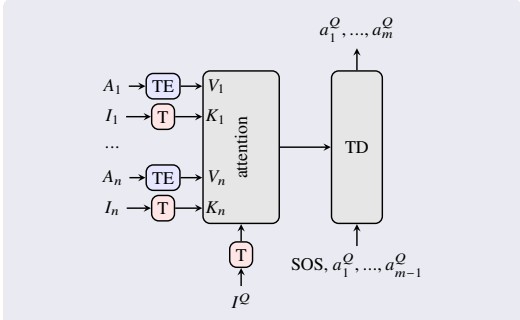

(c) Tree with aligned matrix. A probability tree with an aligned matrix listing the possible outcomes, their probabilities and three columns for events described in later tasks. It uses the grahdrawing library and requires LuaLaTeX.

(d) Our approach is a modified version of **meta-seq2seq**. A transformer decoder (TD) is trained to produce a sequence of actions $a_1^Q, \ldots, a_m^Q$ given a query instruction $I^Q$. The context are demonstrations $(I_k, A_k)$ produced by our generative model. We use a transformer encoder-decoder (T) to encode instructions and state $S$ and a transformer encoder (TE) to encode actions. The transformers that process instructions (pink blocks) receive state $S$ as the input of the encoder.

Figure 5: Illustrative examples from the DATIκZ, licensed for free distribution. The examples are from https://github.com/PetarV-/TikZ, https://github.com/janosh/tikz, https://tikz.net, and https://arxiv.org, respectively.

correlation between the results of the crowdworkers and the experts was strikingly low ($\rho < 0.1$). Since the crowdworkers also showed low agreement among themselves, we decided to discontinue further experiments with crowdsourcing.

# E EXAMPLES

In Figure 5, we provide human-created examples from DATIκZ. Further, Figure 6 contains examples of scientific figures generated by CLIMA$_{13B}$, LLAMA$_{13B}$, and GPT-4 for our human evaluation (cf. §5.2). Each model is represented by one high-scoring and one low-scoring instance, as assessed by our expert annotators in terms of caption similarity. It is worth noting that the low-scoring example of GPT-4 suffers from caption copying. We provide additional examples of caption copying in Figure 7. Instances of generated code can be found in Figure 8.

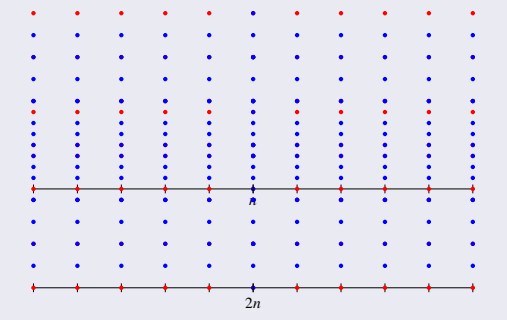
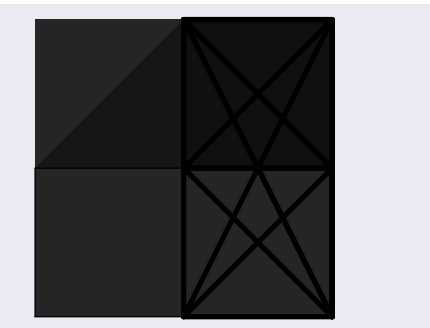

CLIMA$_{13B}$ (good): The ansatz (5.17) and (5.18) for $\alpha = 2$. The red points are evenly spaced, and the blue points scale quadratically with $n$. *The image is a white background with blue and red dots scattered across it. The dots are placed in various arrangements, creating a visually interesting pattern. Some dots are clustered together, while others are spaced further apart, covering the entire background.*

CLIMA$_{13B}$ (bad): For the $P = 2$ scheme, the regular cell footprint is a standard five-point Laplacian, and if any point in the footprint is a cut cell, it is then "irregular." Cut cells are shown with dark shading, irregular cells with light shading, and the remaining white cells are "regular."

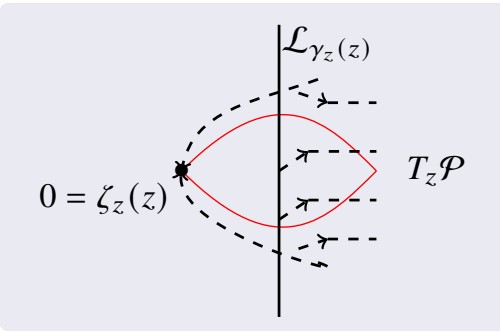
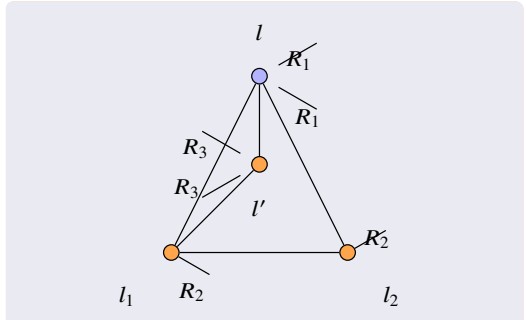

LLAMA$_{13B}$ (good): Local configuration of the path and the foliation (in red) around the point $0 = \Psi_z(z)$. *The image displays a complex mathematical formula with various symbols and notation, including a black dot, waves, and arrows. The formula seems to be related to physics or engineering, and it is written on a white background for easy readability.*

LLAMA$_{13B}$ (bad): Illustration of the proof of Theorem 2.7. *The image displays a tree with blue and orange labels on a white background. The tree has a unique structure, with branches that don't follow a typical tree layout. The labels seem to be representing a formula or a set of instructions, and the tree is accompanied by several equations in the surrounding area.*

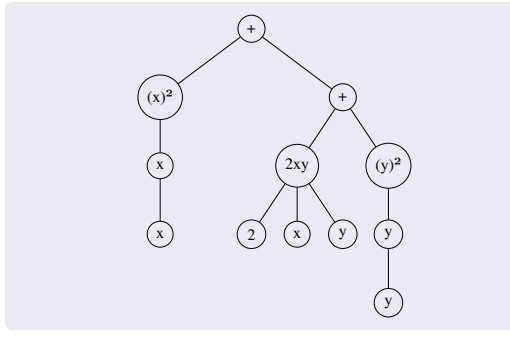
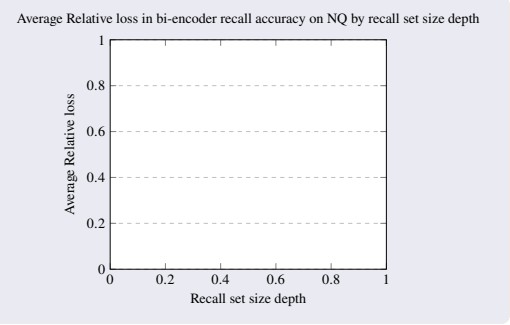

GPT-4 (good): Expression graph for algebraic expression $x^2 + 2xy + y^2$. *The image shows a tree with a symbol at its root, representing a mathematical concept. The tree has a series of logical connections, and there are variables and mathematical symbols throughout the structure. The image conveys a sense of order and organization in the presentation of the mathematical concept.*

GPT-4 (bad): Average Relative loss in bi-encoder recall accuracy on NQ by recall set size depth on the baseline, Pretrained Alignment (PT), Data Augmentation (DA), and Contrastive Alignment Post Training (CAPOT) on noisy queries.

Figure 6: Examples of model-generated scientific figures that received high ratings (good) and low ratings (bad) according to the perception of our expert annotators for caption similarity. The sections of the captions that have been augmented (cf. §3.2) are *emphasized*.

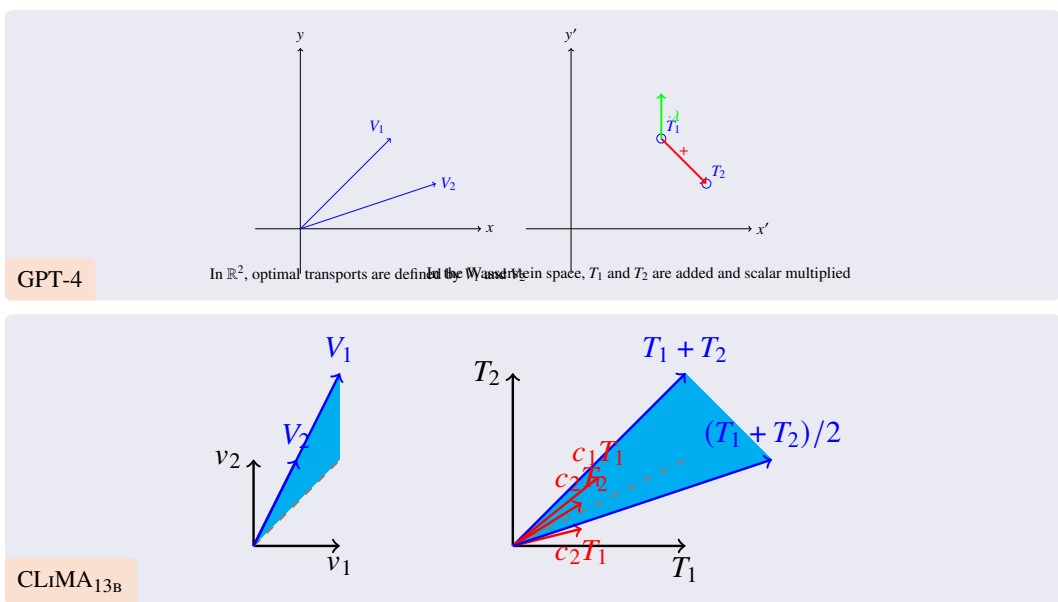

(a) Motivating the definition of the addition ⊕ and scalar multiplication ⊙ operations for $0 < \alpha < 1$ in the Wasserstein optimal transport space for transports $T_1, T_2$ (right), while in $\mathbb{R}^2$ optimal transports are defined by vectors $V_1, V_2$ (left).

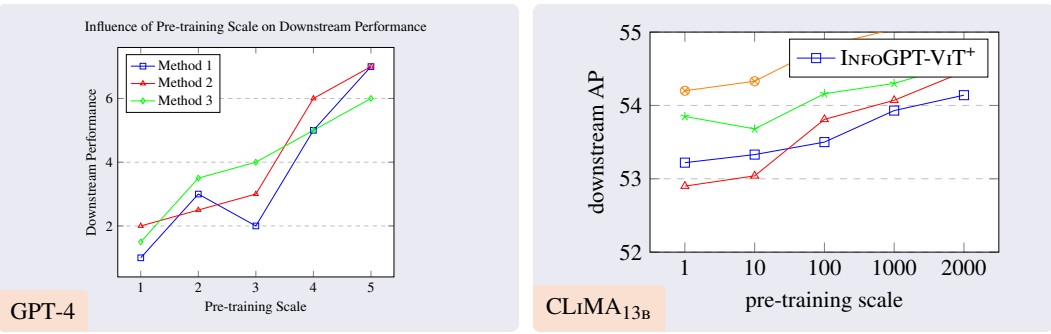

(b) The influence of pre-training scale on the downstream performance. The value of each method is the average metric values across all datasets in the privacy protection setting with LiGHTGBM.

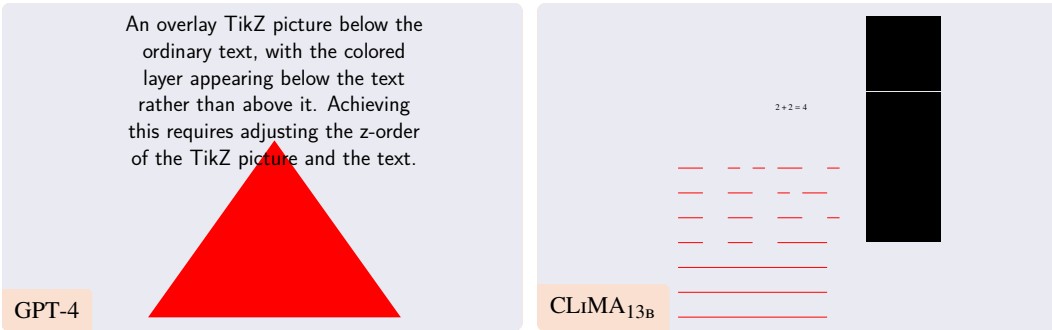

(c) An overlay TikZ picture below the ordinary text, with the colored layer appearing below the text rather than above it. Achieving this requires adjusting the z-order of the TikZ picture and the text.

Figure 7: Additional examples of GPT-4 that suffer from caption copying. For reference, we also provide the output of CLiMA$_{13B}$ for the same captions. The underlying TikZ code of Figure 7c can be found in Figure 8.

```latex
\documentclass{article}
\usepackage{tikz}
\pagestyle{empty}

\begin{document}
\sffamily\Huge
\begin{tikzpicture}[overlay, remember picture]
    \node at (current page.center)
    {
        \begin{tikzpicture}
            \fill[red] (0,0) -- (5,7) -- (10,0) -- cycle;
        \end{tikzpicture}
    };
\end{tikzpicture}

\begin{center}
An overlay TikZ picture below the ordinary text, with the colored layer appearing below
the text rather than above it. Achieving this requires adjusting the z-order of the
TikZ picture and the text.
\end{center}
\end{document}
```

GPT-4: An overlay Ti*k*Z picture below the ordinary text, with the colored layer appearing below the text rather than above it. Achieving this requires adjusting the z-order of the Ti*k*Z picture and the text.

```latex
\documentclass{article}
\usepackage{tikz}
\usepackage{amsmath}
\usetikzlibrary{tikzmark}
\begin{document}
\newcommand{\sep}[1]{\hfill#1\hfill\relax}
\begin{minipage}[t]{.5\linewidth}
\begin{align*}
\sep{2+2=4}
\end{align*}
\end{minipage}%
\begin{minipage}[t]{.5\linewidth}
\tikzmarknode{2}{\rule{3cm}{3cm}}

\tikzmarknode{+}{\rule{3cm}{3cm}}

\tikzmarknode{3}{\rule{3cm}{3cm}}
\end{minipage}
\begin{tikzpicture}[overlay,remember picture]
\draw[red] (-1,-3) -- (5,-3) (-1,-2) -- (5,-2) (-1,-1) -- (5,-1)
(-1,0) -- (0,0) (1,0) -- (2,0) (3,0) -- (5,0)
(-1,1) -- (0,1) (1,1) -- (2,1) (3,1) -- (4,1) (5,1) -- (5.5,1)
(-1,2) -- (0,2) (1,2) -- (2,2) (3,2) -- (3.5,2) (4,2) -- (5,2)
(-1,3) -- (0,3) (1,3) -- (1.5,3) (2,3) -- (2.5,3) (3,3) -- (4,3)
(5,3) -- (5.5,3);
\end{tikzpicture}
\end{document}
```

CLiMA$_{13B}$: An overlay Ti*k*Z picture below the ordinary text, with the colored layer appearing below the text rather than above it. Achieving this requires adjusting the z-order of the Ti*k*Z picture and the text.

Figure 8: The generated code of GPT-4 (top) and CLiMA$_{13B}$ (bottom) from Figure 7c. Both models correctly use the overlay option, but they are not devoid of issues. In particular, GPT-4 utilizes nested tikzpicture environments which is generally discouraged and the output of CLiMA$_{13B}$ seems particularly verbose.

