# OpenReview forum: "AutomaTikZ: Text-Guided Synthesis of Scientific Vector Graphics with TikZ"
_ICLR.cc/2024/Conference — ICLR 2024 poster_

### Official Review · Reviewer_hZYD · 2023-10-30

**Soundness:** 3 good
**Presentation:** 3 good
**Contribution:** 2 fair
**Rating:** 6
**Confidence:** 4

**Summary:**

This paper focuses on TikZ, a language for vector graphics, and provides a dataset, DaTikZ, consisting of 120,000 pairs of TikZ and their captions. The authors are also evaluating the application of LLaMA and CLIP on the dataset. Experimental results show that models trained on DaTikZ can generate more complex vector graphics compared to those synthesized by closed-source LLMs such as GPT-4 and Claude 2.

**Strengths:**

- It is the largest scientific vector image and caption dataset to the best of the reviewer's knowledge.
- Benchmarking with multiple baseline models is reported.
- While automatic evaluation of generative models often encounters difficulties regarding their quality, this paper uses multiple scores for automatic evaluation to make the comparison meaningful. Multiple human subjective evaluations have also been considered and performed, as described in Sections 5.2 and D.

**Weaknesses:**

- As shown in Table 1, DaTikZ consists of multiple data sources. If DaTikZ consists of multiple data sources, the authors should evaluate which data sources contribute to the accuracy of the generation on the test data and to what extent. This would not only justify the use of each data source, but may also suggest what further data should be collected in the future. The contribution of data augmentation should also be evaluated.
- Since this paper is also about a novel dataset for scientific vector graphics and their benchmarking, the technical contribution of the baseline method is modest.

**Questions:**

- The reviewer expects the authors to respond to the points listed in Weaknesses.
- Figure 2 shows two radar charts. Although the main purpose of the radar charts is to make relative comparisons within the same chart, it would be possible to make some comparisons among the method in the left and right charts if the ranges of each score were adjusted.
- When the authors conduct a human annotation campaign, they could also evaluate how well the automatic evaluation metrics used in section 5.1 correlate with subjective evaluation; are there any plans to do so?

---

> ### Author Response · Authors · 2023-11-17
>
> We thank the reviewer for their time and we are happy to respond to their listed weaknesses and questions.
>
> > As shown in Table 1, DaTikZ consists of multiple data sources. If DaTikZ
> > consists of multiple data sources, the authors should evaluate which data
> > sources contribute to the accuracy of the generation on the test data and to
> > what extent. This would not only justify the use of each data source, but may
> > also suggest what further data should be collected in the future.
>
> We agree with the reviewer that this should be investigated. We are currently running these experiments and will report back once we have the results.
>
> > The contribution of data augmentation should also be evaluated.
>
> We are currently in the process of evaluating this. Please refer to our general response for further details.
>
> > Since this paper is also about a novel dataset for scientific vector graphics
> > and their benchmarking, the technical contribution of the baseline method is
> > modest.
>
> We agree with the reviewer that the creation of our dataset plays a central role in our work, however, we believe our methodology carries innovation as well. First, we are, to our best knowledge, the first to combine LLMs with vision encoders to extract visual concepts of *textual data*, which improved CLiMA over LLaMA, despite having the same inputs. This result is by no means trivial. In addition, the multimodality of our data (text/code/images) allows for automated evaluation that is new in this field and lead to new insights. For instance, we identified CLIPScore's vulnerability to typographic attacks and the relative robustness of CLIPScore<sub>img</sub>. Notably, we're perhaps the first to report and use CLIPScore<sub>img</sub> in this work. We have made that more clear in our latest revision.
>
> > Figure 2 shows two radar charts. Although the main purpose of the radar
> > charts is to make relative comparisons within the same chart, it would be
> > possible to make some comparisons among the method in the left and right
> > charts if the ranges of each score were adjusted.
>
> We designed the left radar chart to evaluate our own primary models (architecture & model size), while the right one evaluates other, more heterogeneous models (GPT-4, Claude 2, CLiMA<sub>img</sub>). This leads to more pronounced score differences in the right chart, necessitating different scales. We agree with the reviewer, however, that comparisons across charts would be insightful, which is why we plot the best model from the left chart also in the right one as a reference, and provide exact scores for all models in Appendix C.
>
> > When the authors conduct a human annotation campaign, they could also
> > evaluate how well the automatic evaluation metrics used in section 5.1
> > correlate with subjective evaluation; are there any plans to do so?
>
> Acknowledging the author's feedback, we've updated our paper accordingly and report Spearman correlations for (Caption Similarity-CLIPScore) and (Reference Similarity-CLIPScore<sub>img</sub>). Segment-level correlations are 0.23 and 0.25 (which is similar to what we observe for e.g. machine translation metrics [[1](https://aclanthology.org/2021.tacl-1.87/),[2](https://aclanthology.org/2022.eamt-1.9/)]) and system-level correlations are 0.5 and 0.8, respectively. The higher correlations for image-image over caption-image similarity strengthen our hypothesis regarding CLIPScore's biases in favor of images of texts.

---

> > ### Comment · Reviewer_hZYD · 2023-11-18
> >
> > According to the authors' response, additional reports will be forthcoming in response to the initial review. Correlations between subjective and automatic ratings are also reported (encouraged for inclusion in the paper or supplemental material). The reviewer plans to upgrade the initial score to the accept side depending on the additional reports.

---

> ### Author Response · Authors · 2023-11-22
>
> We are happy to report that we have completed the all ablation studies the reviewer requested and added the results to the newest version of our document.
>
> > As shown in Table 1, DaTikZ consists of multiple data sources. If DaTikZ consists of multiple data sources, the authors should evaluate which data sources contribute to the accuracy of the generation on the test data and to what extent. This would not only justify the use of each data source, but may also suggest what further data should be collected in the future.
>
> We conducted this experiment and found that each data source has primarily positive effects on model performance. When additionally taking the data source sizes into account, we find that `curated examples` have a particularly noticeably positive impact, whereas `TEX Stack Exchange` exhibits smaller effects. For additional details, please refer to Appendix C.3.
>
> > The contribution of data augmentation should also be evaluated.
>
> Please refer to our discussion in the general response section and in Appendix C.3 to see our findings for this ablation experiment.
>
> > Correlations between subjective and automatic ratings are also reported
> > (encouraged for inclusion in the paper or supplemental material).
>
> The requested information can now be found in Appendix C.2 of our re-submission.

---

> > ### Comment · Reviewer_hZYD · 2023-11-23
> >
> > As a result of the authors' previous responses and additional experiments, the reviewer has upgraded the score to the acceptance side of this paper. The datasets provided are found to be conducive to the development of vector graphics generation (albeit with varying degrees of contribution by each dataset). The evaluation metrics are also found to be reasonable and deemed to be conducive to the community as a benchmark.

---

### Official Review · Reviewer_stBp · 2023-10-31

**Soundness:** 2 fair
**Presentation:** 3 good
**Contribution:** 3 good
**Rating:** 6
**Confidence:** 3

**Summary:**

The paper presents AutomaTikZ, a project aimed at automatically generating TikZ drawings from natural language descriptions. The authors introduce a novel dataset, DaTikZ, which consists of aligned TikZ drawings and captions, and a new model architecture named CLiMA that integrates multimodal CLIP embeddings into LLaMA. The results show that CLiMA outperforms both LLaMA and other proprietary models like GPT-4 and Claude 2 in various metrics. CLiMA's capability to process images opens potential applications in vectorization and sketch conditioning.

**Strengths:**

- The paper is well-written and easy to follow.
- The paper introduces a pioneering dataset and a novel model architecture, which is a significant contribution to the field of automatic drawing generation from textual descriptions.
- The proposed model, CLiMA, demonstrates superior performance over existing models, including well-known ones like GPT-4 and Claude 2, across several evaluation metrics.

**Weaknesses:**

- Since the main contribution of the study is introducing DatikZ, it would be better to add a couple of simple captions to the code/image examples in the main draft

**Questions:**

- Can you please provide a couple of examples of different models' behavior to typographic attacks?
- Have others considered comparing with ChatGPT? Maybe fine-tuned on a very small subset of the dataset.

---

> ### Author Response · Authors · 2023-11-17
>
> We thank the reviewer for their time and we are happy to answer their questions.
>
> > Since the main contribution of the study is introducing DatikZ, it would be
> > better to add a couple of simple captions to the code/image examples in the
> > main draft
>
> We are happy to report that we have added additional output results. Please refer to the general response for details.
>
> > Can you please provide a couple of examples of different models' behavior to
> > typographic attacks?
>
> An example of a typographic attack can be seen in Figure 5 (low-right subfigure). We have made that more clear. In our latest revision we have added additional examples. Again, please refer to the general response.
>
> > Have others considered comparing with ChatGPT? Maybe fine-tuned on a very
> > small subset of the dataset.
>
> We have considered ChatGPT in the initial stages of this project, but due to other authors reporting that ChatGPT's performance on generating TikZ [[1](https://arxiv.org/abs/2305.18583), [2](https://arxiv.org/abs/2303.12712)] is vastly below GPT-4, we chose to not include it in our experiments, with GPT-4 effectively serving as an upper bound.

---

### Official Review · Reviewer_L9z3 · 2023-11-01

**Soundness:** 3 good
**Presentation:** 3 good
**Contribution:** 3 good
**Rating:** 8
**Confidence:** 3

**Summary:**

The paper proposes an approach to generating scientific images, by generating their TiKZ code, and open-sources the training dataset for this task. The paper investigates two solutions, namely LLaMa and LLaMa combined with CLIP-based image adapter. Both models are LoRA-tuned. The comparison with promping GPT-4 and Claude-2 is performed.

**Strengths:**

Originality: While this is not the first paper to do vector graphics generation with language models, the specific focus on TikZ is novel, and the published dataset is first of its' kind.

Quality & Clarity: The paper is of high quality, with good related work section, clear metrics,and detailed analysis of the performance, including a human study. It provides the code accompanying the submission, and suggests the availability of the models (link removed for anonymity). The writing is mostly clear, doesn't have typos, the dataset composition is clearly outlined, as are many of the choices made by the authors (such as the choice of LLaMa vs LLaMa-2, the specific prompts used for the models, etc.).

Signficance: The main point of signficance is the release of the dataset, fostering future work in this direction, and I believe it will be useful for the community.

**Weaknesses:**

The main weakness of the paper is the limited ablation study: The authors compare and contrast the two versions of the model, LLaMa and CLiMA (LLaMa + CLIP adapter) but don't highlight other important choices made, ex.
* The effect of the aritifical samples. 50% of the proposed dataset is the artificial samples, but the evaluation is done on real samples. The effect of augmentation with artificial samples is not measured.
* The effect of re-sampling the generation. The authors re-generate the output multiple times until the result is compilable, and the average number of re-generations is > 1.5, but the performance without regeneration is not reported.
* The effect of data augmentation for CLiMa. As specified in section 5, during training CLIP is given either the input caption or the reference image (50-50%) and during inference it is given the input caption. The effect of this data augmentation is not reported.
* The performance of vanilla LLaMa or prompt-tuned LLaMa is not reported, making the comparison between fine-tuned versions and prompted GPT-4 / Claude-2 not exactly fair.

The second issue is the limited model output results reported in the paper - the authors only show 6 examples in the appendix, and out of 3 labeled "good", the first one shows the image that, in my perception, doesn't really match the label ("some dots are clustered together" in the prompt, but shows just rows of points in the image)

**Questions:**

How did the authors arrive at the specific prompt that was used for GPT-4 / Claude-2 and how do they know if that is the one that indeed yields a reasonably high performance from the models?
Given the iterative re-generation, the models could have also made use of the TikZ compiler error / stack trace, what was the rationale for not using this information?

---

> ### Author Response · Authors · 2023-11-17
>
> We thank the reviewer for their time and are pleased to hear that they generally liked our work. We happily answer remaining questions.
>
> > 50% of the proposed dataset is the artificial samples, but the evaluation is
> > done on real samples. The effect of augmentation with artificial samples is
> > not measured.
>
> While the examples are not artificial but augmented, we agree with the reviewer that we should measure this effect. Please refer to our general response for further details.
>
> > The authors re-generate the output multiple times until the result is
> > compilable, and the average number of re-generations is > 1.5, but the
> > performance without regeneration is not reported.
>
> There seems to be a slight misinterpretation regarding the mean number of regenerations; it stands at <=1.2 for all models except Claude, not >1.5 (cf. Table 3). This is a resampling rate we believe is acceptable, especially since resampling is a prerequisite for reporting performance, as the metrics most important to our evaluation all depend on access to compiled images, which is what motivated us to introduce resampling in the first place.
>
> > The effect of data augmentation for CLiMa. As specified in section 5, during
> > training CLIP is given either the input caption or the reference image
> > (50-50%) and during inference it is given the input caption. The effect of
> > this data augmentation is not reported.
>
> Mixing text-only and text-image modalities is common when training models which accept both unimodal and multimodal input [[1](https://arxiv.org/abs/1908.06066),[2](https://arxiv.org/abs/2202.03052)]. However, we do agree with the reviewer that quantifying the effect of this particular augmentation would be insightful. We will add this to our final version.
>
> > The performance of vanilla LLaMa or prompt-tuned LLaMa is not reported,
> > making the comparison between fine-tuned versions and prompted GPT-4 /
> > Claude-2 not exactly fair.
>
> In our preliminary experiments, vanilla LLaMA and its derivatives had negligible TikZ generating capabilities, which is why we did not include them in our comparison. We have cleared this up in our latest revision. We do not think our comparison is necessarily unfair for GPT-4, however. For example, GPT-4 is [rumored](https://news.ycombinator.com/item?id=36675934) to have been trained on joint data (rendered LaTeX/text). In this case, the training would be very similar to our setting and even put our models at a disadvantage, given the likely difference in model size. Either way, since not much is known on how these models operate or how they were trained, we think it is difficult to assess if a comparison is fair or not. What we do show is that our models are a good choice for generating TikZ, outperforming these proprietary models under varying conditions. We have added this to our paper as well.
>
> > The second issue is the limited model output results reported in the paper -
> > the authors only show 6 examples in the appendix
>
> We are happy to report that we have added additional output results. Please refer to the general response for details.
>
> > How did the authors arrive at the specific prompt that was used for GPT-4 /
> > Claude-2 and how do they know if that is the one that indeed yields a
> > reasonably high performance from the models?
>
> We designed our prompts similarly to the ones reported in related work, especially [this one](https://arxiv.org/abs/2305.18583), which arrived at their prompt through prompt engineering. After also experimenting with our own prompts and other prompts stated in [other works](https://arxiv.org/abs/2303.12712), we found the former to work the best in our case. We have made that more clear in our latest version.
>
> > Given the iterative re-generation, the models could have also made use of the
> > TikZ compiler error / stack trace, what was the rationale for not using this
> > information?
>
> Due to GPU constraints, our models were limited to a 1200-token context size. Most examples fit within this size but most were also quite close to the upper limit. Consequently, we cannot fit an additional stack trace and a corrected version within this context. Hence, we offer our iterative resampling technique as a resource-efficient alternative. We have added this information to our latest version.

---

> ### Author Response · Authors · 2023-11-22
>
> We are delighted to provide the outcomes of all ablation studies that the reviewer requested.
>
> > The effect of data augmentation for CLiMa. As specified in section 5, during training CLIP is given either the input caption or the reference image (50-50%) and during inference it is given the input caption. The effect of this data augmentation is not reported.
>
> In response, we conducted an ablation experiment and have included our positive findings in Appendix C.3. This data augmentation indeed positively influenced the model's performance, as we initially hypothesized.
>
> > 50% of the proposed dataset is the artificial samples, but the evaluation is done on real samples. The effect of augmentation with artificial samples is not measured.
>
> Please refer to our second general response and Appendix C.3 for a comprehensive discussion of our findings from this ablation experiment.

---

> > ### Comment · Reviewer_L9z3 · 2023-11-22
> > **Reply to authors**
> >
> > I thank the authors for answering my questions and their updates to the manuscript. I am keeping my rating.

---

### Official Review · Reviewer_1jqj · 2023-11-03

**Soundness:** 3 good
**Presentation:** 3 good
**Contribution:** 3 good
**Rating:** 6
**Confidence:** 4

**Summary:**

In this paper, authors collect 119k TiKz datat to fine-tune LLaMa, and achieve better performance compared to GPT4 and Claude2 in TiKz generation.
Also, authors propose a method to prevent the TiKz code from not compiling.

**Strengths:**

1. Good approach error handling method. Quite interesting study: on a specific domain, with relatively small data scale (119k) and LoRA, the author's approach can actually perform pretty well.
2. Evaluation metrics are extensive and conving.
3. Presentation is clear.

**Weaknesses:**

1. Seems like author didn't show any conversation example where users can edit or optimize the code while in the chat.
2. Recently, [1] propose to leverage SVG, a similar format to TiKz, to conduct image understanding, generation and editing. How do authors judge on this?
3. The error handling strategy seems quite standard and wildly used in programming languages.

[1] Cai, Mu, Zeyi Huang, Yuheng Li, Haohan Wang, and Yong Jae Lee. "Leveraging Large Language Models for Scalable Vector Graphics-Driven Image Understanding." arXiv preprint arXiv:2306.06094 (2023).

**Questions:**

1. Any code error rate comparsion? (compiling success rate)

---

> ### Author Response · Authors · 2023-11-17
>
> We thank the reviewer for their time and we are happy to answer their questions.
>
> > Seems like author didn't show any conversation example where users can edit
> > or optimize the code while in the chat.
>
> While we agree that this would be an interesting direction for future work, we want to remark that our models are not general-purpose, instruction-tuned models, i.e., it is not possible to chat with them. We emphasize this more in our latest revision. This deliberate design choice is akin to text-to-image models like Stable Diffusion to which we draw parallels in this work.
>
> > Recently, [1] propose to leverage SVG, a similar format to TikZ, to conduct
> > image understanding, generation and editing. How do authors judge on this?
>
> We thank the author for providing this new paper. We think that when it comes to scientific figures, it has similar limitations to other works that generate SVGs which we discuss in the introduction and related work sections, and which motivated us to use TikZ in the first place. We added this paper to our related work section in our latest reversion.
>
> > The error handling strategy seems quite standard and wildly used in
> > programming languages.
>
> Since most related works are about high-resource programming languages, they can leverage the much richer available tooling to prevent errors. Consequently, we are not aware of any other works which employ our error mitigating technique designed for low-resource settings. Therefore, we do not think it is as a standard technique and we believe it is a meaningful contribution.
>
> > Any code error rate comparison? (compiling success rate)
>
> In our experiments, the amount of errors the models make is directly proportional to the compilation success rate. We added this information to our latest reversion.

---

### Author Response · Authors · 2023-11-17
**General Response**

We would like to thank all reviewers for their valuable comments. Based on their feedback, we have updated our documents main body and appendices, highlighting the relevant changes in blue.

As a response to the collective request to see more examples, we gladly report that we have included examples from our DaTikZ dataset in the appendix, and further increased the examples showcasing caption copying.

Remarks by Reviewer 2 & 4 surrounding a deeper analysis on our data augmentation impact remain in our attention. While we show in Section 3.2  that this augmentation improves the alignment of image-caption pairs, we agree that we should also evaluate the direct effect on our models. We are currently running these experiments and intend to revise our submission again once the results are available.

---

> ### Author Response · Authors · 2023-11-22
>
> We are pleased to state that we have successfully completed all of the requested ablation experiments. We have analyzed the effect of data augmentation of captions (as requested by reviewers 3 & 4) using CLiMA-7b as a representative model for efficiency reasons and our discoveries support our initial intuition: data augmentation of image captions has a predominantly positive effect on model performance. Please see the newly added Appendix C.3 in our updated manuscript for further details on this subject. For other individually requested ablation experiments, see our revised responses below within each section.

---

### Meta-Review · Area_Chair_TcXf · 2023-12-06

**Metareview:**

The paper "AutomaTikZ", aiming to automate TikZ drawings generation from natural language descriptions, has been thoroughly reviewed by the reviewers, each highlighting its strengths and pointing out areas for improvement. The reviewers collectively appreciated the paper's originality, particularly its focus on TikZ and the introduction of the novel DaTikZ dataset. The paper's soundness, presentation, and contributions were generally rated good. Key strengths identified include the novelty of the dataset, clarity and quality of the paper, and the paper's potential to foster future work in vector graphics generation. The methodology involving LoRA-tuning of LLaMa and CLiMA (LLaMa combined with CLIP-based image adapter) and their comparison with GPT-4 and Claude-2 was also commended.

However, reviewers noted several weaknesses. These include a lack of user interaction examples, limited ablation studies, and potential gaps in the comparative analysis. Some reviewers suggested more extensive discussion on topics like SVG-based methods, the effect of artificial samples and data augmentation in the dataset, and the performance of vanilla LLaMa. Questions about the specific prompts used for GPT-4 and Claude-2 and the rationale for certain methodological choices were raised. The authors effectively addressed these concerns in their responses, clarifying methodological decisions, adding results and examples to the paper, and conducting additional experiments to cover the gaps identified. This thorough and responsive approach led to a positive reassessment of the paper, with reviewers acknowledging the effort and upgrading their ratings, ultimately leaning towards acceptance of the paper.

**Justification For Why Not Higher Score:**

"AutomaTikZ" is rated "accept as poster" instead of the higher score as the paper relatively lacks the extensive depth or breadth typically suitable for the spotlight or oral category.

**Justification For Why Not Lower Score:**

The paper was unanimously accepted by the reviewers. No basis to overturn the reviews.

---

### Decision · Program_Chairs · 2024-01-16

Accept (poster)